# Emigration and fiscal gap in population-exporting region

**Mei-Qi Li**[1]*, **Yong Zhang**[2]

**1** School of Taxation, Jilin University of Finance and Economics, Changchun, Jilin, P. R. China, **2** School of Electronic Information Engineering, Changchun University of Science and Technology, Changchun, Jilin, P. R. China

* mqli1809@163.com

## Abstract

This paper analyzes how emigration impacts fiscal gap of population-exporting region in the long term. We construct a general equilibrium model of emigration and fiscal gap and make empirical verification using two-step system GMM model. Among the major lessons from this work, five general and striking results are worth highlighting: (1) the economic losses of emigration are the immediate cause of widening the fiscal gap. (2) in the short and long term, emigration can expand the fiscal revenue gap through the superimposed effect of tax rate and tax base. (3) the gap in fiscal expenditure is widened by the outflow of people in the short term. However, local governments would change the strategy to keep the spending gap from widening in the long run. (4) a positive impact of emigration on the fiscal gap. the more severe population emigration, the larger the fiscal gap. (5) when the trend of emigration becomes irreversible, the subsequent efforts of local governments to expand fiscal expenditure for attraction population would not only fail to revive the regional economy, but aggravate the expansion of fiscal gap. The contribution of research is twofold. On the one hand, it fills the theoretical gap between emigration and fiscal gap because previous studies have paid little attention to the fiscal problems of local government of population outflow. On the other hand, the selection of Northeast China that has been subject to long-term out-of-population migration is good evidence to verify this theory, which is tested very well using the 2S-GMM model. The comprehensive discussion on the relationship between emigration and fiscal gap is helpful to guide those continuous population-exporting regions that are facing a huge fiscal gap how to solve the fiscal gap and unsustainability from the perspective of fiscal revenue and expenditure.

**Data Availability Statement:** All relevant data are within the manuscript and its Supporting Information files.The original data can be found in the website(https://data.cnki.net/yearBook?type=type&code=A).

## Introduction

In developing countries, due to the different speed of economic development among cities, the income gap and living cost gap of people are constantly widening, which leads to enormous gaps of GDP among cities. And then affects the fiscal decisions of local governments and the change of specific fiscal revenue and expenditure. In a short period of time, the population-exporting regions have to face the lack of labor supply and the decline in investment because

**Funding:** This work was supported by General Project of Education Department of Jilin Province in China under Grant No. JJKH20230167SK, General Project of Jilin University of Finance and Economics in China under Grant No. 2022YB027, and Science and Technology Development Plan Project of Jilin Province in China under Grant No. 20210101078JC. All funders of the funding projects are the authors of this paper.

**Competing interests:** The authors have declared that no competing interests exist.

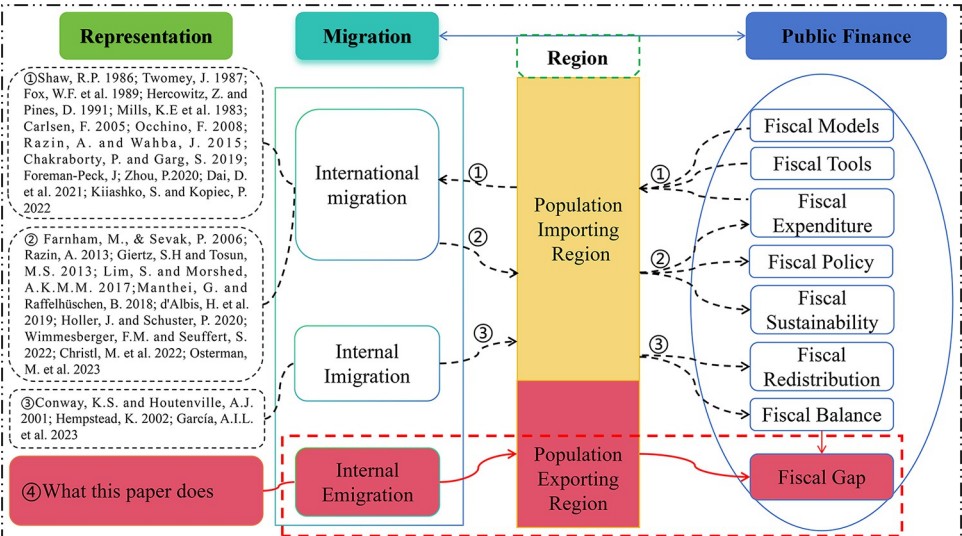

**Fig 1. The literature review of the issue of migration and public finance.**

the people who are attracted to work and settle in the rapidly developing areas leave their original places of residence [1,2]. More seriously, the local governments of the population-exporting regions have to face a huge fiscal gap, which is caused by insufficient fiscal revenue and huge fiscal expenditure pressure. In a large period of time, the massive emigration has led to a long-term fiscal unsustainability of the local governments who are at higher risk of developing fiscal and economic crisis. Thus, it is essential to cause alarm and attention to the population-exporting regions which is the vulnerable groups. And this question doesn't get much attention in research. In this paper, we should build a theoretical model to explain the impact mechanism of population migration on fiscal gap of population-exporting regions and find empirical evidence for verification, which can not only supplement the theoretical system of migration and finance, but also help governments find strategies to ensure that fiscal gap be not widened and fiscal sustainability is maintained.

The literature review of the issue of migration and public finance is shown in Fig 1. Research on the subject of migration and public finance has been growing since the beginning of the migration phenomenon, which is carried out from the following three aspects. First, the impact of local fiscal conditions on immigration decisions. It is included the impact of fiscal models on immigration decisions [3]; The impact of fiscal structure on immigration decisions [4,5]; The effect of the host's fiscal situation on the migration outcome [6]; Using fiscal tools to attract immigrants [7–9], such as the use of tax policies to attract immigrants [9,10] and generate fiscal competition [11,12]; And the impact of public services on immigration [13]. Second, the impact of migration on the public finance in receiving economies [14–16]. It is included the impact of migration on public services [17]; The impact of immigration on fiscal costs and benefits [18]; The impact of immigration on fiscal revenues [19,20]; The impact of migrant remittances on fiscal policy [21,22]; The impact of migration on the welfare of groups in receiving countries [23] (Holler J, Schuster P. 2020); And the impact of migration on the long-term fiscal sustainability in receiving countries [24,25]. Third, the impact of internal migration on public finance [26]. The research on migration is no longer limited to international migration, but pays more attention to internal and inter-regional migration. It is included the impact of fiscal redistributive spending on regions [27]; The impact of immigration on tax incentives in exporting regions [28].

Drawing from the literature review on migration and public finance, the shortage of existing researches are as follows: First, under the topic of the relationship between migration and public finance, it is paid more attention to immigration and the financial situation of the economies that receive them. There are relatively few studies on the impact of emigration on the fiscal situation of domestic population-exporting region. We can see that people seem to miss the point that the losses suffered by the economies that send migrants are not the same as the benefits gained by the economies that receive them. This neglect has led to a blind eye to the damage of migration to the economic and fiscal balance of population-exporting region. Thus, this topic should require in-depth scholarly attention which is exactly what this paper does. Second, most of the research have not fully discussed the fiscal part, which are usually from the perspective of the relationship between fiscal expenditure and immigration, or between fiscal policy and immigration. This paper would comprehensively analyze the relationship between the fiscal revenue and expenditure of a region of emigration, in order to fill the gap of existing research.

We conduct the research from two parts: theoretical model and empirical analysis. The graphical abstract is shown in S1 Fig in S1 Appendix. We construct a general equilibrium model of migration and fiscal gap, which interpret the effect of population emigration on the different kinds of fiscal gaps (tax revenue and government expenditure) in outflowing regions and get three added values: (1) the economic losses of emigration are the immediate cause of widening the fiscal gap in the migrant-exporting regions. (2) in the short and long term, population emigration can expand the fiscal revenue gap through the superimposed effect of tax rate and tax base. (3) the gap in fiscal expenditure is widened by the outflow of people in the short term. However, local government would change the strategy to keep the spending gap from widening in the long run through cutting maintenance-oriented spending.

In the part of empirical analysis, we chose 36 cities of the Northeast China as the research object, where are serious population emigration and receive a high proportion of general transfer payments from the central government. The perfect empirical results are obtained on the two-step GMM system, and get two added values: (4) a positive impact of population emigration on the fiscal gap in population-exporting region. the more severe population emigration, the larger the fiscal gap. (5) when the trend of emigration becomes irreversible, the subsequent efforts of local governments to expand fiscal expenditure for attraction population would not only fail to revive the regional economy, but aggravate the expansion of the regional fiscal gap.

The contribution of research is twofold. On the one hand, it fills the theoretical gap between emigration and fiscal gap because previous studies have paid little attention to the fiscal problems of local government of population outflow. On the other hand, the selection of an area that has been subject to long-term out-of-population migration is good evidence to verify this theory, which is tested very well using the 2S-GMM model. The comprehensive discussion on the relationship between emigration and fiscal gap is helpful to guide those continuous population-exporting regions that are facing a huge fiscal gap how to solve the fiscal gap problem from the perspective of fiscal revenue and expenditure.

The paper structure is as follows. After introduction, the part of Theoretical Mechanism Analysis has constructed a general equilibrium model of emigration and fiscal gap. The part of Data and Methodology has presented the data and methodology of the empirical analysis. The part of Empirical Results and Discussion has analyzed the empirical results of two-step system GMM approach. The part of Conclusions, as the last part, has summarized the main conclusions and prospected the future research direction.

## Theoretical mechanism analysis

The issue that local governments can attract population inflow through setting tax policies and allocating fiscal expenditure has attracted considerable attention from scholars in recent decades. However, there is not yet a definitive theoretical mechanism to explain the fiscal gap of outflowing areas who are affected by population migration. By referring to the fiscal balance model [29,30], this paper constructs a model of emigration and fiscal gap to study the influencing mechanism of emigration on the fiscal revenue and expenditure in the outflow areas.

Considering that the fiscal gap of governments refers to the difference between fiscal revenue and expenditure. If the fiscal expenditure is more (or less) than the revenue, it would be a fiscal deficit (or surplus). For the local governments of population outflow, the fiscal gap is manifested as the fiscal expenditure is greater than the fiscal revenue. The general government fiscal gap ($FP_t$) is defined as:

$$FP_t = \sum_i FR_{i,t} - \sum_i FE_{i,t} - PD_{t-1} \qquad (2-1)$$

where $FR_{i,t}$ is local government fiscal revenue from tax revenue and nontax revenue in the current period, $FE_{i,t}$ is the primary expenditures$\in$ {purchase expenditure and transfer expenditure} in the current period. $PD_{t-1}$ is local government gross debt in the previous period. Nevertheless, local debt does not change significantly with emigration. It is not a core explanatory variable in following models. Therefore, the Hypothesis 1 of the model is as follows:

**Hypothesis 1**: The balance of government debts is as a constant in the model.

In Eq (2-1), both sides are divided by nominal output ($Y_t^N$) which can be rewritten as:

$$fp_t = \sum_i fr_{i,t} - \sum_i fe_{i,t} - pd_{t-1} \qquad (2-2)$$

where $fp_t = \frac{GFP_t}{Y_t^N}$, $fr_{i,t} = \frac{FR_{i,t}}{Y_t^N}$, $fe_{i,t} = \frac{FE_{i,t}}{Y_t^N}$, and $pd_{t-1} = \frac{PD_{t-1}}{Y_t^N}$. Combined with macroeconomic function (Y = C + I + G + NX), Eq (2-3) is formed by population as a factor in regional economic development:

$$Y_{i,t} = Y_{i,t}(C_{i,t}(L_{i,t}), I_{i,t}(L_{i,t}), G_{i,t}(L_{i,t}), NX_t) \qquad (2-3)$$

where $Y_{i,t}$ represents the level of economic development in $i$ area, $C_{i,t}$ is the power of consumption in $i$ area, which is assumed to vary automatically with labor ($L_{i,t}$), $I_{i,t}$ is the power of investment in $i$ area, which is assumed to vary automatically with labor ($L_{i,t}$), $G_{i,t}$ is the level of local government purchase in $i$ area, which is assumed to vary automatically with labor ($L_{i,t}$), $NX_t$ is the level of net exports in $i$ area. We assume that net exports have inconspicuous effect on the change of local population., and $L_{i,t}$ is the scale of labor in $i$ area. A first-order Taylor expansion of $Y_{i,t}$ to $L_{i,t}$ yields the following:

$$Y_t - Y_i^* = \left[ \frac{\partial Y_{i,t}}{\partial C_{i,t}} \frac{\partial C_{i,t}}{\partial L_{i,t}} + \frac{\partial Y_{i,t}}{\partial I_{i,t}} \frac{\partial I_{i,t}}{\partial L_{i,t}} + \frac{\partial Y_{i,t}}{\partial G_{i,t}} \frac{\partial G_{i,t}}{\partial L_{i,t}} \right] (L_t - L^*) \qquad (2-4)$$

where $\frac{\partial Y_{i,t}}{\partial C_{i,t}}$, $\frac{\partial Y_{i,t}}{\partial I_{i,t}}$ and $\frac{\partial Y_{i,t}}{\partial G_{i,t}}$ are the partial derivative of *GDP* of local government with respect to the consumption, investment and government purchase, respectively, $\frac{\partial C_{i,t}}{\partial L_{i,t}}$ is the partial derivative of consumption with respect to the labor, where $\frac{\partial I_{i,t}}{\partial L_{i,t}}$ is the partial derivative of investment with respect to the labor, where $\frac{\partial G_{i,t}}{\partial L_{i,t}}$ is the partial derivative of government purchase with respect to

the labor. The function formed by labor is shown in Eq (2-5):

$$L_{i,t} = L_{i,t}(H_t, R_t, IMR_t, NPR_t) \qquad (2-5)$$

where $H_t$ represents the population in the base period, $R_t$ represents the labor participation rate, $IMR_t$ is the net migration rate, and $NPR_t$ is the natural population growth rate.

## Emigration and the gap of fiscal revenue

According to the classification of fiscal revenue, it is including tax revenue ($TR_{i,t}$) and non-tax revenue ($NTR_{i,t}$). Tax revenue is affected by the tax base ($TB_{i,t}$) and the tax rate ($r_{i,t}$). Based on past experiences, there is a direct relationship between nontax revenue and tax revenue in China, that the tax revenue of local governments is getting more, while the non-tax revenue is getting less. We can identify the non-tax revenue as a part of fiscal revenue related to taxation. Therefore, the discussion of fiscal revenue only establishes the equation from the perspective of taxation and the Hypothesis 2 of the model is as follows:

**Hypothesis 2:** The non-tax revenue as the part of as a part of fiscal revenue related to taxation is a constant in the model.

The Eq (2-6) can be written as follows:

$$FR_{i,t} = FR_{i,t}(TR_{i,t}, NTR_{i,t}) \approx TR_{i,t}(TB_{i,t}(Y_{i,t}, \tau_{i,t}(Y_{i,t})), \tau_{i,t}(Y_{i,t})) \qquad (2-6)$$

A first-order Taylor expansion of government tax yields the follows:

$$FR_{i,t} - FR_i^* = \left[ \frac{\partial FR_{i,t}}{\partial TR_{i,t}} \frac{\partial TR_{i,t}}{\partial TB_{i,t}} \frac{\partial TB_{i,t}}{\partial Y_{i,t}} \frac{\partial Y_{i,t}}{\partial L_{i,t}} + \frac{\partial FR_{i,t}}{\partial TR_{i,t}} \frac{\partial TR_{i,t}}{\partial \tau_{i,t}} \frac{\partial \tau_{i,t}}{\partial Y_{i,t}} \frac{\partial Y_{i,t}}{\partial L_{i,t}} + \frac{\partial FR_{i,t}}{\partial TR_{i,t}} \frac{\partial TR_{i,t}}{\partial TB_{i,t}} \frac{\partial TB_{i,t}}{\partial \tau_{i,t}} \frac{\partial \tau_{i,t}}{\partial Y_{i,t}} \frac{\partial Y_{i,t}}{\partial L_{i,t}} \right]$$
$$\times (Y_t - Y^*) \qquad (2-7)$$

where $\frac{\partial FR_{i,t}}{\partial TR_{i,t}}$ is the partial derivative of fiscal revenue with respect to the tax revenue, $\frac{\partial TR_{i,t}}{\partial TB_{i,t}}$ is the partial derivative of tax revenue with respect to the tax base, $\frac{\partial TB_{i,t}}{\partial Y_{i,t}}$ is the partial derivative of tax base with respect to GDP, $\frac{\partial Y_{i,t}}{\partial L_{i,t}}$ is the partial derivative of GDP with respect to the labor, $\frac{\partial TR_{i,t}}{\partial \tau_{i,t}}$ is the partial derivative of tax revenue with respect to the tax rate, $\frac{\partial TB_{i,t}}{\partial \tau_{i,t}}$ is the partial derivative of tax base with respect to the tax rate, $\frac{\partial \tau_{i,t}}{\partial Y_{i,t}}$ is the partial derivative of tax rate with respect to GDP. Multiplying in the both side of the Eq (2-7) by $\frac{1}{Y^*}$, and then multiplying the right side of Eq (2-7) by 1, yields the following fiscal revenue gap:

$$\frac{FR_{i,t} - FR_i^*}{Y^*} = \frac{FR_i^*}{Y^*} \varepsilon_{Y_{i,t}, L_{i,t}} \left( \varepsilon_{FR_i, TR_i} \varepsilon_{TR_i, TB_i} \varepsilon_{TB_i, Y_{i,t}} + \varepsilon_{FR_i, TR_i} \varepsilon_{TR_i \tau_i} \varepsilon_{\tau_i, Y_{i,t}} + \varepsilon_{FR_i, TR_i} \varepsilon_{TR_i, TB_i} \varepsilon_{TB_i, \tau_i} \varepsilon_{\tau_i, Y_{i,t}} \right) \bar{y}_t \quad (2-8)$$

where $\bar{y}_t$ is the GDP gap, $\varepsilon_{Y_{i,t}, L_{i,t}}$ is the elasticity of GDP with respect to the labor, $\varepsilon_{FR_i, TR_i}$ is the elasticity of fiscal revenue with respect to the tax revenue, $\varepsilon_{TR_i, TB_i}$ is the elasticity of tax revenue with respect to the tax base, $\varepsilon_{TB_i, Y_{i,t}}$ is the elasticity of tax base with respect to GDP, $\varepsilon_{TR_i \tau_i}$ is the elasticity of tax revenue with respect to the tax rate, $\varepsilon_{\tau_i, Y_{i,t}}$ is the elasticity of tax rate with respect to GDP, $\varepsilon_{TR_i \tau_i}$ is the elasticity of tax revenue with respect to the tax rate.

According to Eqs (2-4), (2-5) and (2-8), we can elucidate that population combined with technology and capital constitute human capital and private investment (or savings), respectively. For regions with large population loss, the young and middle-aged people, the vast majority of them are high-level talents with higher education and high technical level, can affect the technological innovation ability of local industry, the production capacity of local industry and the per capita income level of the region. The decline in consumption is

inevitable. Meanwhile, under the premise that local labor demand remains unchanged in the short term, we can suppose that population outflow leads to a decrease in supply of labor market. In the wake of the relative increase in the demand for labor, the demand price in the labor market would rise due to the decrease scale and the change of age structure in the labor force in the long term. As a result, the decline of investment in industry and services is appear, as well as the scale of $Y_t$. By $\varepsilon_{Y_{i,t},L_{i,t}}$, we can observe that the GDP gap gets bigger with the decrease of population scale and the change of labor force structure ($L_{i,t}$). Through $\varepsilon_{TB_i,Y_{i,t}}$, $\varepsilon_{\tau_i,Y_{i,t}}$ and $\varepsilon_{TR_i\tau_i}$ superimposed on the GDP, the gap of fiscal revenue is more prominent.

## Emigration and the gap of fiscal expenditure

According to the classification of the economic nature of expenditure, the structure of fiscal expenditure that is assumed to vary automatically with GDP ($Y_{i,t}$) can be divided into two categories: government purchase expenditure ($PS_{i,t}$) and government transfer expenditure ($TS_{i,t}$). Furthermore, government purchasing expenditure can be subdivided into two parts, one is the expenditure to maintain normal operation of local governments, which in this paper is called $PS_{i,t}^a$. The other is the expenditure closely related to people's daily production and life, such as education, environmental protection, culture, science and technology, which is called purchase spending ($PS_{i,t}^b$). This type of the spending is assumed to vary automatically with the fiscal policy ($P_t^d$) related to public demand. Government transfer expenditure is assumed to vary automatically with the fiscal policy ($P_t^s$) related to the local fiscal policy, which is including social security and employment expenditures, and health care expenditures. The specific division of the structure on fiscal expenditure is shown in Table 1 according to the source of data. The Hypothesis 3 of the model is as follows:

**Hypothesis 3:** The division of government expenditure types is related to the fiscal objectives achieved by the local governments.

The Eq (2-9) can be written as follows:

$$FE_{i,t} = FE_{i,t}(PS_{i,t}, TS_{i,t}) = FE_{i,t}(PS_{i,t}^a(Y_{i,t}), PS_{i,t}^b(Y_{i,t}, P_t^d(Y_{i,t})), TS_{i,t}(Y_{i,t}, P_t^s(Y_{i,t}))) \qquad (2-9)$$

A first-order Taylor expansion of government spending yields the follows:

$$FE_{i,t} - FE_i^* = \left[ \frac{\partial FE_{i,t}}{\partial PS_{i,t}^a} \frac{\partial PS_{i,t}^a}{\partial Y_{i,t}} \frac{\partial Y_{i,t}}{\partial L_{i,t}} + \frac{\partial FE_{i,t}}{\partial PS_{i,t}^b} \frac{\partial PS_{i,t}^b}{\partial Y_{i,t}} \frac{\partial Y_{i,t}}{\partial L_{i,t}} + \frac{\partial FE_{i,t}}{\partial TS_{i,t}} \frac{\partial TS_{i,t}}{\partial Y_{i,t}} \frac{\partial Y_{i,t}}{\partial L_{i,t}} + \frac{\partial FE_{i,t}}{\partial PS_{i,t}^b} \frac{\partial PS_{i,t}^b}{\partial P_t^d} \frac{\partial P_t^d}{\partial Y_{i,t}} \frac{\partial Y_{i,t}}{\partial L_{i,t}} + \frac{\partial FE_{i,t}}{\partial TS_{i,t}} \frac{\partial TS_{i,t}}{\partial P_t^s} \frac{\partial P_t^s}{\partial Y_{i,t}} \frac{\partial Y_{i,t}}{\partial L_{i,t}} \right] (Y_t - Y^*) \quad (2-10)$$

where $\frac{\partial FE_{i,t}}{\partial PS_{i,t}^a}$ is the partial derivative of fiscal expenditure with respect to the purchase spending

**Table 1. The specific division of the structure on fiscal expenditure.**

| Group | Classification |
|---|---|
| **Purchase Spending (a)** | General Public Services |
| | Public Safety |
| | Urban and Rural Community Management |
| **Purchase Spending (b)** | Education |
| | Culture, Sports and Media |
| | Transportation |
| | Agriculture, Forestry and Water Management |
| | Environmental Protection |
| | Science and Technology |
| **Transfer Spending** | Social Security and Employment |
| | Medical and Healthcare |

Note: In this table, the categories are divided according to the way of fiscal expenditures in China.

(a), $\frac{\partial FE_{i,t}}{\partial PS^b_{i,t}}$ is the partial derivative of fiscal expenditure with respect to the purchase spending (b),

$\frac{\partial FE_{i,t}}{\partial TS_{i,t}}$ is the partial derivative of fiscal expenditure with respect to the transfer spending, $\frac{\partial PS^b_{i,t}}{\partial P^d_t}$ is the

partial derivative of the purchase spending (b) with respect to the fiscal policy, $\frac{\partial TS_{i,t}}{\partial P^s_t}$ is the partial

derivative of the transfer spending (b) with respect to the fiscal policy, $\frac{\partial PS^a_{i,t}}{\partial Y_{i,t}}$ is the partial derivative

of the purchase spending (a) with respect to GDP, $\frac{\partial PS^b_{i,t}}{\partial Y_{i,t}}$ is the partial derivative of the purchase

spending (b) with respect to GDP, $\frac{\partial TS_{i,t}}{\partial Y_{i,t}}$ is the partial derivative of transfer spending with respect to

GDP. $\frac{\partial TS_{i,t}}{\partial P_t}$ is the partial derivative of transfer spending with respect to fiscal policy. $\frac{\partial P_t}{\partial Y_{i,t}}$ is the partial

derivative of fiscal policy with respect to GDP. Multiplying in the both side of the Eq (2-10) by $\frac{1}{Y^*}$,

and then multiplying the right side of Eq (2-10) by 1, yields the following fiscal expenditure gap:

$$\frac{FE_{i,t} - FE^*_i}{Y^*} = \frac{FE^*_i}{Y^*} \varepsilon_{Y_{i,t},L_{i,t}} \big( \varepsilon_{FE_i,PS^a_{i,t}} \varepsilon_{PS^a_{i,t},Y_{i,t}} + \varepsilon_{FE_i,PS^b_{i,t}} \varepsilon_{PS^b_{i,t},Y_{i,t}} + \varepsilon_{FE_i,TS_{i,t}} \varepsilon_{TS_{i,t},Y_{i,t}} + \varepsilon_{FE_i,PS^b_{i,t}} \varepsilon_{PS^b_{i,t},P^d_t} \varepsilon_{P^d_t,Y_{i,t}}$$
$$+ \varepsilon_{FE_i,TS_{i,t}} \varepsilon_{TS_{i,t},P^s_t} \varepsilon_{P^s_t,Y_{i,t}} \big) \bar{y}_t \qquad (2-11)$$

where $\varepsilon_{FE_i,PS^a_{i,t}}$ is the elasticity of fiscal expenditure with respect to the purchase spending (a),

$\varepsilon_{FE_i,PS^b_{i,t}}$ is the elasticity of fiscal expenditure with respect to the purchase spending (a), $\varepsilon_{FE_i,TS_{i,t}}$ is

the elasticity of fiscal expenditure with respect to the purchase spending (b), $\varepsilon_{PS^b_{i,t},P^d_t}$ is the elasticity

of the purchase spending (b) with respect to the fiscal policy, $\varepsilon_{TS_{i,t},P^s_t}$ is the elasticity of the transfer

spending with respect to the fiscal policy, $\varepsilon_{PS^a_{i,t},Y_t}$ is the elasticity of purchase spending (a) with

respect to GDP, $\varepsilon_{PS^b_{i,t},Y_t}$ is the elasticity of purchase spending (b) with respect to GDP, $\varepsilon_{TS_{i,t},Y_t}$ is the

elasticity of transfer spending with respect to GDP.

According to Eqs (2-4), (2-5) and (2-11), we can give a interpret that government purchasing expenditure is mainly invested in the early stage and maintained in the later stage so that it is difficult to really reduce the financial burden of the government on emigration in the short term. For example, local governments cannot eliminate administrative offices and fire employees because of the annual exodus. It is necessary to establish indispensable administrative structures in each city. Therefore, $PS^a_{i,t}$ cannot show a downward trend in the short term. In the long run, the governments should cancel or merge some unnecessary administrative agencies and reduce the corresponding fiscal expenditure due to the continuous emigration. In order to retain population and attract new ones, the scale of $PS^b_{i,t}$ based on GDP can increase. $TS_{i,t}$ accounts for a higher proportion of the overall fiscal expenditure structure because of the larger share of the non-labor population. In the long term, this situation can crowd out the share of $PS_{i,t}$, so that $TS_{i,t}$ become larger.

### Emigration and fiscal gap

Fig 2 shows the impact of emigration on the fiscal gap. According to, Eqs (2-8) and (2-11), Eq (2-2) can be rewritten:

$$fp_t = fp^* + e_{yl}(\theta \bar{y}_t + \rho \bar{y}_t) - pd_{t-1} \qquad (2-12)$$

where $fp^* = \sum_i \left[ \frac{FR^*_i - FE^*_i}{Y^*} \right]$ is the fiscal balance in the state of equilibrium, $e_{yl} = \varepsilon_{Y_{i,t},L_{i,t}}$ is the elas-

ticity of GDP with respect to the fiscal balance, $\theta = \sum_i \left[ \frac{FR^*_i}{Y^*} \left( \varepsilon_{FR_i,TR_i} \varepsilon_{TR_i,TB_i} \varepsilon_{TB_i,Y_{i,t}} + \varepsilon_{FR_i,TR_i} \varepsilon_{TR_i \tau_i} \right. \right.$

$\varepsilon_{\tau_i, Y_{i,t}} \big) - \frac{FE^*_i}{Y^*} \left( \varepsilon_{FE_i,PS^a_{i,t}} \varepsilon_{PS^a_{i,t},Y_t} + \varepsilon_{FE_i,PS^b_{i,t}} \varepsilon_{PS^b_{i,t},Y_t} + \varepsilon_{FE_i,TS_{i,t}} \varepsilon_{TS_{i,t},Y_t} \right) \Big]$ is the public budget elasticity,

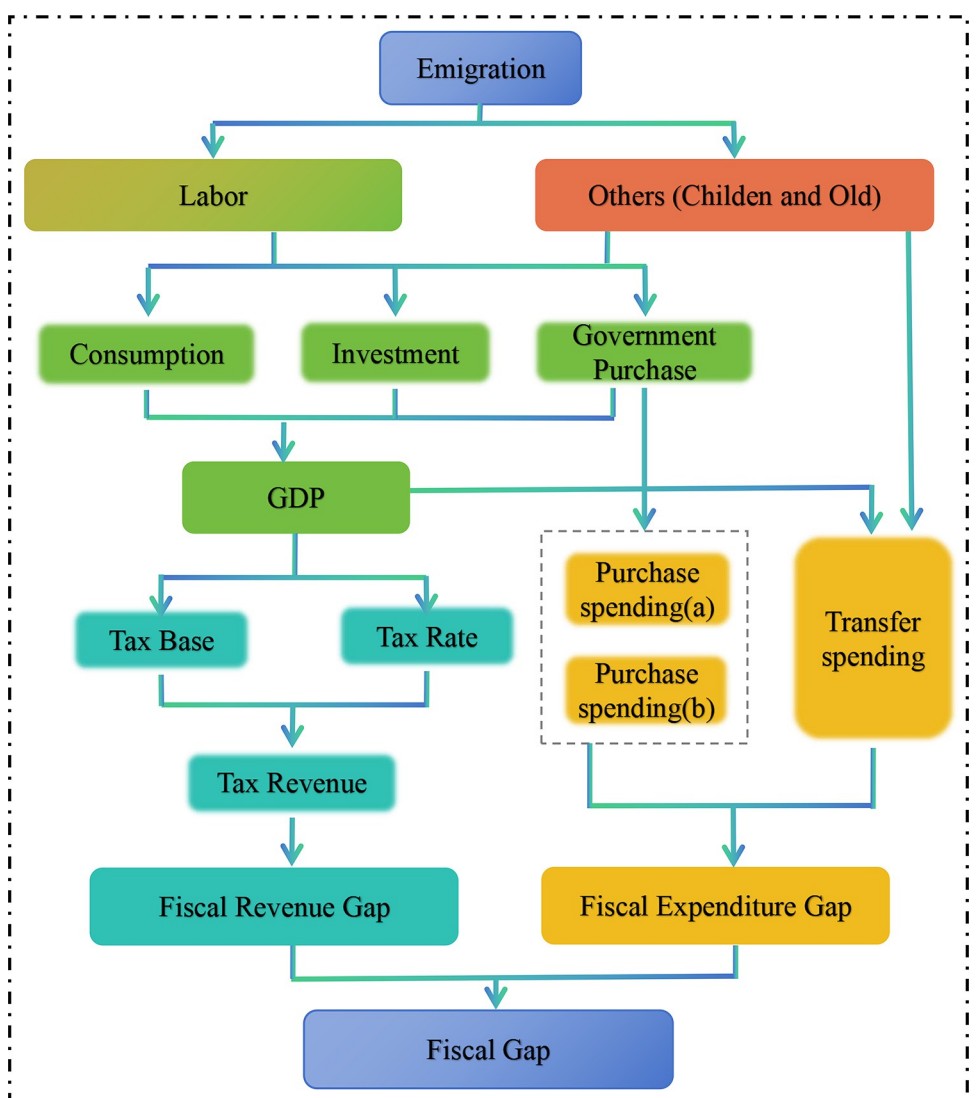

**Fig 2. The impact of emigration on the fiscal gap.**

$\rho = \sum_i \left[ \frac{FR_i^*}{Y^*} \boldsymbol{\varepsilon}_{FR_i,TR_i} \boldsymbol{\varepsilon}_{TR_i \tau_i} \boldsymbol{\varepsilon}_{\tau_i,Y_{i,t}} - \boldsymbol{\varepsilon}_{FE_i,PS_{i,t}^b} \boldsymbol{\varepsilon}_{PS_{i,t}^b,P_t^d} \boldsymbol{\varepsilon}_{P_t^d,Y_{i,t}} - \boldsymbol{\varepsilon}_{FE_i,TS_{i,t}} \boldsymbol{\varepsilon}_{TS_{i,t},P_t^s} \boldsymbol{\varepsilon}_{P_t^s,Y_{i,t}} \right]$ is the public budget elasticity for the fiscal policy of the central and local governments. Eq (2-12) shows that the fiscal gap is generated through the GDP gap and the change of fiscal policy under the impact of population emigration.

## Data and methodology

### Data

The regional scope of traditional Northeast China was divided into two senses that were narrow sense and broad sense. First, the narrow sense referred to Liaoning Province, Jilin Province and Heilongjiang Province. Second, the broad sense referred to the three northeast provinces and the four cities (leagues) in the east of the Inner Mongolia Autonomous Region. The narrow sense was selected as the research sample owing to the availability and

completeness of data. The research sample includes 14 of cities in Liaoning Province, 9 of cities (autonomous prefectures) in Jilin Province, and 13 of cities (administrative office) in Heilongjiang Province.

The original data of the research sample derived from the following: 'Liaoning Statistical Yearbook', 'Jilin Statistical Yearbook', and 'Heilongjiang Statistical Yearbook' from 2009 to 2021; statistical yearbook of various cities in Northeast China from 2009 to 2021; and the annual statistical bulletin of national economic and social development of various cities in Northeast China from 2010 to 2020.

## Methodology

In the empirical analysis, the choice of model was considered from two aspects. On the one hand, the time dimension was small and the individual cross-section dimension was large in the panel data. On the other hand, the multiple test results were judged after regression. Eventually, system GMM was chosen to achieve the empirical analysis [31–33]. The general form of the regression equation is as follows:

$$y_{i,t} = \gamma + \alpha_1 y_{i,t-1} + \alpha_2 y_{i,t-2} + \cdots + \alpha_p y_{i,t-p} + \boldsymbol{\beta x}_{i,t} + \boldsymbol{\delta}_i Control\ variables_{i,t} + u_i + \varepsilon_i$$

In this equation, $y_{i,t}$ represented the explained variables, $y_{i,t-p}$ represented P-order lag term of the explained variables, $x_{i,t}$ represented the explanatory variables, $z_i$ represented the time invariant individual characteristics, $u_i + \varepsilon_i$ represented the compound random disturbance term, $\gamma$ represented the constant term, $\alpha_p$ represented the coefficient of the lag term of explained variables, $\beta$ represented the coefficient of explanatory variables, $\delta_i$ represented the coefficient of individual characteristics.

## Definition and calculation of variables

The calculation method for each variable is shown in Table 2 and the descriptive statistics of all variables are shown in Table 3. S2–S13 Figs in S1 Appendix show the data of all variables of 36 cities in Northeast China, which represent the accuracy of data and the individual differences and commonalities of research samples. As mentioned above, a total of six models are constructed, corresponding to six explained variables, which are: Fiscal revenue gap (*FR*), Fiscal expenditure gap (*FE*), Purchase spending gap (*PSa*), Purchase spending gap (*PSb*), Transfer spending gap (*TS*), and Fiscal Gap (*FG*). The explanatory variable is Population emigration (*PE*). Fig 3 shows that the fitting relation between the explained variables and the explanatory variable, respectively. The control variables are GDP Growth (*GDP*), Average Income (*PCI*), Open (*Open*), Invest Growth (*Invest*), Population Growth (*PG*). Before building models, correlation tests between each set of variables are shown in Fig 4, which can verify whether the selected indicators are suitable. The results of correlation tests are shown in S1 Table in S1 Appendix.

According to the general expression of system GMM and the selected variables, the empirical models of population emigration on fiscal gap focus on the establishment of systematic GMM models. Due to the inertia of fiscal gap, the current period of fiscal gap was affected by the previous period. Therefore, the first-order lag terms of the explained variables were added into the models to build dynamic panel models. The specific model Eqs (3-1)–(3-6) are as follows:

$$FR_{i,t} = \gamma + \alpha FR_{i,t-1} + \beta PE_{i,t} + \delta_i Control\ variables_{i,t} + u_i + \varepsilon_i \tag{3-1}$$

$$FE_{i,t} = \gamma + \alpha FE_{i,t-1} + \beta PE_{i,t} + \delta_i Control\ variables_{i,t} + u_i + \varepsilon_i \tag{3-2}$$

**Table 2. The calculation methods for each variable.**

| Variable Label | Variable | Calculation Method |
|---|---|---|
| **Explained Variables** | FR | $\frac{fiscal\ revenue_n - fiscal\ revenue_{n-1}}{fiscal\ revenue_{n-1}}$ |
| | FE | $\frac{fiscal\ expenditure_n - fiscal\ expenditure_{n-1}}{fiscal\ expenditure_{n-1}}$ |
| | PSa | $\frac{purchase\ spending(a)_n - purchase\ spending(a)_{n-1}}{purchase\ spending(a)_{n-1}}$ |
| | PSb | $\frac{purchase\ spending(b)_n - purchase\ spending(b)_{n-1}}{purchase\ spending(b)_{n-1}}$ |
| | TS | $\frac{transfer\ spending_n - transfer\ spending_{n-1}}{transfer\ spending_{n-1}}$ |
| | FG | $fiscal\ gap_n = fiscal\ revenue_n - fiscal\ expenditure_n$ |
| | | $FG_n = \frac{fiscal\ gap_n - fiscal\ gap_{n-1}}{fiscal\ gap_{n-1}}$ |
| **Explanatory Variable** | PE | $\frac{regsident\ population_n - houseehold\ population_n}{regsident\ population_n}$ |
| **Control Variables** | GDP | $\frac{GDP_n - GDP_{n-1}}{GDP_{n-1}}$ |
| | PCI | $\frac{GDP_n}{resident\ population_n}$ |
| | Open | $open_n = export_n + import_n$ |
| | | $Open = \frac{open_n - open_{n-1}}{open_{n-1}}$ |
| | Invest | $\frac{fixed\ assets\ investment_n - fixed\ assets\ investment_{n-1}}{fixed\ assets\ investment_{n-1}}$ |
| | PG | $birth\ rate_n - death\ rate_n$ |

Note: The select of calculation method should take into consideration the availability of data and the need of empirical analysis.

$$PSa_{i,t} = \gamma + \alpha PSa_{i,t-1} + \beta PE_{i,t} + \delta_i Control\ variables_{i,t} + u_i + \varepsilon_i \qquad (3-3)$$

$$PSb_{i,t} = \gamma + \alpha PSb_{i,t-1} + \beta PE_{i,t} + \delta_i Control\ variables_{i,t} + u_i + \varepsilon_i \qquad (3-4)$$

$$TS_{i,t} = \gamma + \alpha TS_{i,t-1} + \beta PE_{i,t} + \delta_i Control\ variables_{i,t} + u_i + \varepsilon_i \qquad (3-5)$$

$$FG_{i,t} = \gamma + \alpha FG_{i,t-1} + \beta PE_{i,t} + \delta_i Control\ variables_{i,t} + u_i + \varepsilon_i \qquad (3-6)$$

**Table 3. Descriptive statistics of all variables.**

| Variable | N | Mean | SD | Min | Max |
|---|---|---|---|---|---|
| FR | 396 | 0.085 | 0.209 | -0.586 | 0.856 |
| FE | 396 | 0.105 | 0.117 | -0.288 | 0.690 |
| PSa | 396 | 0.090 | 0.193 | -0.674 | 1.066 |
| PSb | 396 | 0.114 | 0.197 | -0.600 | 1.387 |
| TS | 396 | 0.129 | 0.206 | -0.361 | 3.018 |
| FG | 396 | 0.132 | 0.198 | -0.671 | 1.663 |
| PE | 396 | -0.066 | 0.113 | -0.766 | 0.193 |
| GDP | 396 | 0.046 | 0.131 | -0.469 | 0.368 |
| PCI | 396 | 4.507 | 2.190 | 1.354 | 14.297 |
| Open | 396 | 0.135 | 1.266 | -0.987 | 23.785 |
| Invest | 396 | 0.047 | 0.323 | -0.915 | 3.884 |
| PG | 396 | -1.553 | 3.971 | -16.500 | 5.900 |

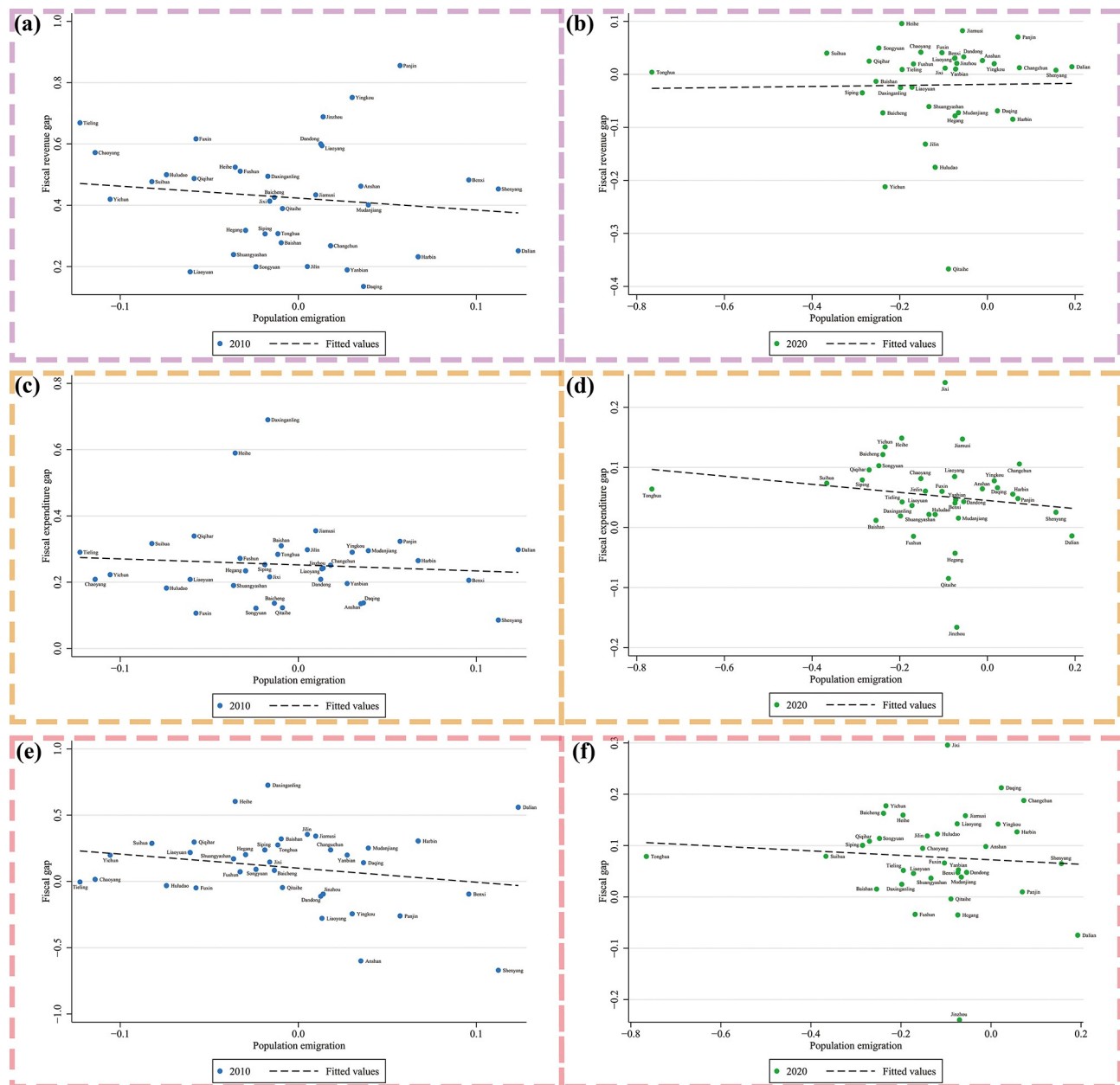

**Fig 3. The fitting relation of fiscal gap and population emigration.** Fig (a) and Fig (b) show the fitting relationship between fiscal revenue gap and population emigration of 36 cities in Northeast China in 2010 and 2020, respectively. Fig (c) and Fig (d) show the fitting relationship between fiscal expenditure gap and population emigration of 36 cities in Northeast China in 2010 and 2020, respectively. Fig (e) and Fig (f) show the fitting relationship between fiscal gap and population emigration of 36 cities in Northeast China in 2010 and 2020, respectively.

## Empirical results and discussion

### Stationarity tests

**Unit root tests.** The results of unit root tests are shown in Table 4. The methods of unit root tests have included LLC, IPS, Fisher-ADF and Fisher-PP. All variables strongly have rejected the null hypothesis of panel unit root, except for PE and Invest by the above four test methods. Subsequently, the first-order difference tests are performed. The results of tests show

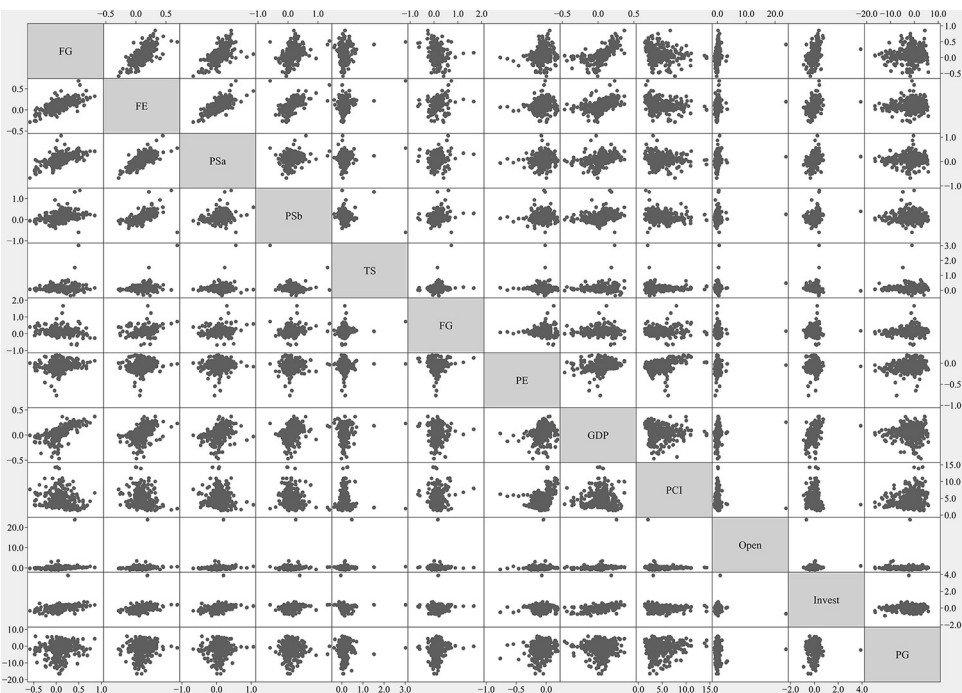

**Fig 4. The correlation tests between each set of variables.** The correlation test refers to the Pearson correlation coefficient, which is used to describe linear correlation strength between variables. The value is between -1 and 1. When two variables have a strong linear correlation, the correlation coefficient is close to 1 or -1. Under the total normal distribution hypothesis, a set of variables can define a correlation matrix after the correlation coefficients are defined for two variables. The X-axis and Y-axis show the range ability of the correlation coefficients of each group of variables.

that all variables have rejected the null hypothesis of panel unit root and kept the same order stationary. In other words, all variables after the first-order difference are stationary in the unit root tests. The results of unit root tests should be used to determine the necessity of cointegration tests.

**Cointegration tests.** Although the results of unit root tests are stable after the first-order difference of all variables, the economic meanings of variables are different from the original sequences. In this paper, the original sequence regressions are necessary to better present the economic relationship between the explained variables and the explanatory variables. Hence, cointegration tests should be done to verify whether there is a long-term equilibrium relationship between variables after unit root tests. The methods of cointegration tests have included Kao-ADF tests [34] and Pedroni tests [35,36]. The Kao-ADF tests have five types of t-value that are less than 0.0001, while the Pedroni tests have three types of t-value that are less than 0.0001. All above the results of cointegration tests represent the inexistence of cointegration relationship between variables. In other words, there is a long-term equilibrium relationship between the explained variables and the explanatory variables in all models. The results of the cointegration tests are shown in Table 5 that the relationships between the explained variables and the explanatory variables are represented by Variable Relationship (1)–(6).

## Model estimation and result analysis

According to the results of the stationarity tests, the regression estimate can be continued. Comparative experiments and specific comparative results are shown in S2 Table in

**Table 4. Unit root tests.**

| Variables | Fisher-ADF | LLC | Fisher-PP | IPS | Unit root |
|---|---|---|---|---|---|
| *FR* | 9.690 | -11.981 | 11.248 | -4.336 | Yes |
| | (0.000) | (0.000) | (0.000) | (0.000) | |
| *FE* | 11.287 | -11.755 | 19.567 | -5.948 | Yes |
| | (0.000) | (0.000) | (0.000) | (0.000) | |
| *PSa* | 5.621 | -6.963 | 16.657 | -6.684 | Yes |
| | (0.000) | (0.000) | (0.000) | (0.000) | |
| *PSb* | 5.664 | -7.483 | 36.028 | -7.434 | Yes |
| | (0.000) | (0.000) | (0.000) | (0.000) | |
| *TS* | 14.013 | -6.434 | 64.234 | -9.430 | Yes |
| | (0.000) | (0.000) | (0.000) | (0.000) | |
| *FG* | 24.224 | -14.379 | 46.378 | -9.129 | Yes |
| | (0.000) | (0.000) | (0.000) | (0.000) | |
| *PE* | 6.679 | 69.297 | 5.107 | 0.188 | **NO** |
| | (0.000) | (1.000) | (0.000) | (0.575) | |
| *D.PE* | 1.198 | 39.723 | 47.354 | -7.084 | Yes |
| | (0.115) | (1.000) | (0.000) | (0.000) | |
| *GDP* | 8.219 | -12.721 | 3.991 | -4.065 | Yes |
| | (0.000) | (0.000) | (0.000) | (0.000) | |
| *PCI* | 4.342 | -6.249 | 8.902 | -2.508 | Yes |
| | (0.852) | (0.000) | (0.000) | (0.006) | |
| *Open* | 14.149 | -21.560 | 18.381 | -6.328 | Yes |
| | (0.000) | (0.000) | (0.000) | (0.000) | |
| *Invest* | 0.5995 | -4.262 | 21.309 | -5.946 | **NO** |
| | (0.274) | (0.000) | (0.000) | (0.000) | |
| *D. Invest* | 13.862 | -9.820 | 59.191 | -8.347 | Yes |
| | (0.000) | (0.000) | (0.000) | (0.000) | |
| *PG* | 4.529 | -6.888 | 21.289 | -7.784 | Yes |
| | (0.000) | (0.000) | (0.000) | (0.000) | |

Note: In this table, the result of Fisher-ADF and Fisher-PP is the value of Modifed inv. Chi-squared Pm. The result of LLC is the value of adjusted t, and the result of IPS is the value of Z-t-tilde-bar. The Yes or No indicates the existence or inexistence of unit root.

S1 Appendix. Thus, S-GMM is used to estimate model (1)–(6), correspondingly the model Eqs (3-1)–(3-6). The dynamic panel GMM estimation can be divided into one-step estimation and two-step estimation according to the different weight matrices. The two-step system GMM method is chosen in order to ensure the satisfaction of three conditions. First, the standard error presents obvious decreasing bias. Second, the perturbation term is not autocorrelated. Third, the moment condition is not overly constrained.

At the same time, in order to ensure the effectiveness of system GMM, two tests are selected to identify the effectiveness of the instrumental variables and the estimated results. First, the Arellano-Bond test method is used to verify the first-order autocorrelation and second-order autocorrelation for the first-difference disturbance terms. It can be found that the first-difference terms of model (1)–(6) have the first-order autocorrelation. Nevertheless, the second-order autocorrelation are inexistence. According to the Arellano-Bond test, six equations are effective because it accepts the original assumption that the disturbance terms have no autocorrelation. Second, the over identifying tests are used to determine whether the instrumental variables selections are excessive under the GMM estimation method. There are two test

**Table 5. Cointegration tests.**

| Variable Relationship | Kao-ADF tests | | | | Pedroni tests | | |
|---|---|---|---|---|---|---|---|
| | MDF | DF | ADF | UMDF | MPP | PP | ADF |
| **(1)** | -5.373 | -11.623 | -6.179 | -13.314 | 9.828 | -7.989 | -5.296 |
| | (0.000) | (0.000) | (0.000) | (0.000) | (0.000) | (0.000) | (0.000) |
| **(2)** | -7.016 | -14.499 | -7.932 | -14.804 | 9.745 | -9.129 | -7.910 |
| | (0.000) | (0.000) | (0.000) | (0.000) | (0.000) | (0.000) | (0.000) |
| **(3)** | -7.457 | -14.554 | -4.748 | -16.806 | 9.571 | -10.531 | -8.922 |
| | (0.000) | (0.000) | (0.000) | (0.000) | (0.000) | (0.000) | (0.000) |
| **(4)** | -7.164 | -15.627 | -11.055 | -15.964 | 9.987 | -11.648 | -9.140 |
| | (0.000) | (0.000) | (0.000) | (0.000) | (0.000) | (0.000) | (0.000) |
| **(5)** | -8.685 | -21.745 | -6.655 | -16.894 | 9.300 | -17.320 | -13.422 |
| | (0.000) | (0.000) | (0.000) | (0.000) | (0.000) | (0.000) | (0.000) |
| **(6)** | -10.123 | -19.487 | -9.714 | -16.400 | 9.856 | -12.191 | -11.238 |
| | (0.000) | (0.000) | (0.000) | (0.000) | (0.000) | (0.000) | (0.000) |

Note: In this table, P-value in parentheses. The cointegration test of Kao has five results of t-value, which is Modified Dickey-Fuller t (MDF), Dickey-Fuller t (DF), Augmented Dickey-Fuller t (ADF), Unadjusted modified Dickey-Fuller t (UMDF) and Unadjusted Dickey-Fuller t (UDF). We choose the first four t-values to account for the variable relationship. The cointegration test of Pedroni has three results, which is Modified Phillips-Perron t (MPP), Phillips-Perron t (PP) and Augmented Dickey-Fuller t (ADF).

methods including Hansen test and Sargan test. Sargan tests are selected because Hansen tests are prone to too many instrumental variables and the results of Hansen tests are not significant. Through Sargan tests, the results can be found that the null hypothesis of all instrumental variables are valid is accepted, indicating that the model setups can be used for system GMM estimation. The P-values of Wald chi-2 test are all less than 0.001, and the P-values of all variables are all less than 0.100 in the model regression results, indicating that the models have good fitting effects.

Therefore, the results of the model estimation are reliable. The results of the model estimation are shown in Table 6. The coefficients of the first order lag of the six explained variables are significant at the significance level of 1%, indicating the fiscal revenue and expenditure of the previous period can affect the fiscal revenue and expenditure of the current period. The coefficients of Population Outflow are significantly negative, indicating that the scale of population outflow is very unfavorable for the fiscal balance.

## The impact of population emigration on fiscal revenue gap

The empirical results display that the population emigration has a positive impact on the fiscal revenue gap of Northeast China. It means that more and more people have left their residence so that the fiscal revenue gap of local governments have been increased. The gap of fiscal revenue increases 0.512 units because of the net population emigration increased by every unit. Under the current tax rate and tax base, the consequence of emigration would continue to worsen the situation of local fiscal revenue. It has to find some new ways if the local governments want to increase the local fiscal revenue. In the past 10 years, many local governments in Northeast China believed that the income from land usage right and taxes brought by the real estate industry is the effective way to maintain local fiscal revenue. However, with the accumulation of bubbles in the real estate industry, it is difficult for this way to have positive effects in the future. Among the control variables, expect for PCI, control variables have positive influence on the fiscal revenue. Thereinto, GDP Growth that added every unit increases

**Table 6. Model estimation results.**

| Variables | Model (1) | Model (2) | Model (3) | Model (4) | Model (5) | Model (6) |
|---|---|---|---|---|---|---|
| L. FR | 0.253*** | | | | | |
| | (10.40) | | | | | |
| L. FE | | 0.098*** | | | | |
| | | (3.42) | | | | |
| L. PSa | | | -0.257*** | | | |
| | | | (-6.19) | | | |
| L. PSb | | | | -0.256*** | | |
| | | | | (-21.52) | | |
| L. TS | | | | | -0.206*** | |
| | | | | | (-14.54) | |
| L. FG | | | | | | -0.029** |
| | | | | | | (-2.07) |
| PE | 0.512*** | 0.238*** | 0.243** | 0.211** | 0.184*** | 0.152 |
| | (2.66) | (3.50) | (2.49) | (2.02) | (4.11) | (1.55) |
| GDP | 0.307*** | 0.122*** | 0.498*** | 0.231*** | 0.120*** | -0.151*** |
| | (7.73) | (5.38) | (8.12) | (6.73) | (7.44) | (-5.28) |
| PCI | -0.108*** | -0.049*** | -0.064*** | -0.066*** | -0.019*** | 0.067*** |
| | (-11.43) | (-16.77) | (-10.43) | (-10.57) | (-6.46) | (18.97) |
| Open | 0.022*** | 0.011*** | 0.020*** | 0.006** | 0.024*** | 0.012*** |
| | (5.68) | (4.36) | (6.62) | (2.23) | (12.68) | (3.81) |
| Invest | 0.049*** | 0.080*** | 0.108*** | 0.175*** | -0.074*** | 0.076*** |
| | (4.11) | (7.03) | (7.66) | (11.94) | (-24.56) | (6.75) |
| PG | 0.004*** | 0.001 | 0.004*** | -0.006*** | 0.003*** | 0.002 |
| | (4.71) | (1.06) | (4.25) | (-5.47) | (6.53) | (1.59) |
| Constant | 0.548*** | 0.317*** | 0.389*** | 0.428*** | 0.256*** | -0.149*** |
| | (15.56) | (19.98) | (12.76) | (21.93) | (17.77) | (-6.72) |
| AR (1) | 0.0000 | 0.0003 | 0.0000 | 0.0006 | 0.0013 | 0.0016 |
| AR (2) | 0.5379 | 0.3908 | 0.5583 | 0.2032 | 0.7670 | 0.8289 |
| Sargan | 0.1622 | 0.1551 | 0.1858 | 0.1116 | 0.1034 | 0.1590 |
| Wald | 0.0000*** | 0.0000*** | 0.0000*** | 0.0000*** | 0.0000*** | 0.0000*** |

Note:

*** $p < 0.01$

** $p < 0.05$

* $p < 0.1$, Z-value in parentheses, Arellano-Bond tests for AR (1) and AR (2), Sargan tests for Sargan, and Wald tests for Wald.

the fiscal revenue by 0.307 units, which is the biggest impact among variables. The control variables of Open and Invest is the driving force of economic development, which leads to the increase in the fiscal revenue. The positive influence of Open and Invest on the fiscal revenue are expressed as 0.071. It means that investment and foreign trade that added every unit increases the growth rate of fiscal revenue by 0.071 units. Moreover, the net benefit of population growth to fiscal revenues is weak. The fiscal revenue can be increased by 0.004 units merely because of the net growth of population in Northeast China increased by 1 unit.

## The impact of population emigration on fiscal expenditure gap

The population emigration has a positive impact on the scale and the structure of the fiscal expenditure gap in Northeast China. It means that more and more people have left their

residence so that the fiscal expenditure of local governments have been decreased. The scale of the fiscal expenditure gap enlarges 0.238 units because of the net population outflow increased by every unit. It can be found that the impact of population emigration on the fiscal revenue gap is greater than that on the scale of the fiscal expenditure gap at the same time. In the short run, there has been no significant reduction in government spending to offset the decline in fiscal revenue. it would become more serious in the long run if this situation is not adjusted in the government budget. The fiscal sustainability will continue to deteriorate.

In the three parts of the fiscal expenditure, the positive impact of population emigration on the purchase spending (a), purchase spending (b) and transfer spending are 0.243, 0.211 and 0.184, respectively. The combined effect of population emigration on the three parts is much larger than the effect of population emigration on the total fiscal spending, that the offsetting effects between the three fiscal spending may weaken the overall effect. The impact of the population outflow on purchase spending (a) is the biggest. With the continuous population emigration, the demand and supply for the various of public goods that are exclusive and competitive have reduced in Northeast China. The government administration spending is the primary part of the purchase spending (a), which increases with the increased scale of the effective resident in the jurisdiction. But it is not cut down significantly with the increased population outflow.

## The impact of population emigration on fiscal gap

The population emigration has a positive impact on the fiscal gap in Northeast China. It means that more and more people have left their residence so that the fiscal balance of local governments have been worse. The fiscal gap enlarges 0.152 units because of the net population emigration increased by every unit. It indicates that the scale of population emigration in Northeast China is greater, the fiscal self-sufficiency is worse, and transfer payments and tax rebates given by the central government to support the financial operation are greater. It is noteworthy that GDP has the negative impact to the fiscal gap, which means GDP that added every unit decreases the growth rate of fiscal gap by 0.151 units. The result verifies the results of the general equilibrium model, which is GDP is the most significant affecting factor of fiscal gap among many control variables.

## Conclusion

In conclusion, the following conclusions are obtained. We have explored the relationship between emigration and fiscal gap from the general equilibrium model. It needs to be investigated the changes in the fiscal gap over time. On the one hand, In the short and long term, emigration can expand the fiscal revenue gap through the superimposed effect of tax rate and tax base. It has to find some new ways if the local governments want to increase the local fiscal revenue, such as introducing new local taxes or non-tax revenue. On the other hand, the exodus widens the fiscal spending gap in the short term while the spending gap would continue to widen in the long run if the local governments do not change their strategies. In the structure of fiscal expenditure, the government purchasing expenditure is mainly invested in the early stage and maintained in the later stage so that it is difficult to really reduce the financial burden of the government on emigration in the short term. Meanwhile, in order to retain population and attract new ones, the scale of purchase spending based on GDP would increase. Due to the structure of local population, the proportion of transfer expenditure would be relatively high. In the empirical analysis, we have taken 36 cities in Northeast China as the research samples, which have faced the serious population emigration in recent ten years, to verify the general equilibrium model. The emigration has a positive impact on the fiscal gap, that the more severe

population emigration, the larger the fiscal gap. When the trend of emigration becomes irreversible, the subsequent efforts of local governments to expand fiscal expenditure for attraction population would not only fail to revive the regional economy, but aggravate the expansion of fiscal gap.

This work fills the theoretical gap between migration and public finance, and its conclusions can help ameliorate fiscal unsustainability. Future studies can continue to explore the economic gains of population-receiving areas and the economic losses of population-exporting areas. We can find a suitable combination of fiscal policies to reduce the efficiency losses. We can consider that the fiscal balance of these population-exporting regions can be better solved from the perspective of space.

## Supporting information

**S1 Appendix.**
(DOCX)

## Author Contributions

**Conceptualization:** Mei-Qi Li.

**Data curation:** Mei-Qi Li.

**Formal analysis:** Mei-Qi Li.

**Funding acquisition:** Mei-Qi Li, Yong Zhang.

**Investigation:** Mei-Qi Li.

**Methodology:** Mei-Qi Li.

**Resources:** Mei-Qi Li.

**Supervision:** Mei-Qi Li.

**Validation:** Mei-Qi Li.

**Writing – original draft:** Mei-Qi Li.

**Writing – review & editing:** Mei-Qi Li, Yong Zhang.

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
