## [Decision Letter · Decision Letter 0]

5 Mar 2024

PONE-D-24-02684Emigration and fiscal gap in population-exporting regionPLOS ONE

Dear Dr. Li,

Thank you for submitting your manuscript to PLOS ONE. After careful consideration, we feel that it has merit but does not fully meet PLOS ONE’s publication criteria as it currently stands. Therefore, we invite you to submit a revised version of the manuscript that addresses the points raised during the review process.

**ACADEMIC EDITOR: The reviewers suggested minor revision so carefully incorporate the comments.** 

We look forward to receiving your revised manuscript.

Kind regards,

Ghulam Rasool Madni, Ph.D

Academic Editor

PLOS ONE

3. We notice that your supplementary tables are included in the manuscript file. Please remove them and upload them with the file type 'Supporting Information'. Please ensure that each Supporting Information file has a legend listed in the manuscript after the references list.

4. Please remove your figures from within your manuscript file, leaving only the individual TIFF/EPS image files, uploaded separately. These will be automatically included in the reviewers’ PDF.

Reviewers' comments:

Reviewer's Responses to Questions

**Comments to the Author**

1. Is the manuscript technically sound, and do the data support the conclusions?

Reviewer #1: Yes

Reviewer #2: Yes

2. Has the statistical analysis been performed appropriately and rigorously? 

Reviewer #1: Yes

Reviewer #2: Yes

3. Have the authors made all data underlying the findings in their manuscript fully available?

Reviewer #1: Yes

Reviewer #2: Yes

4. Is the manuscript presented in an intelligible fashion and written in standard English?

Reviewer #1: Yes

Reviewer #2: Yes

5. Review Comments to the Author

Reviewer #1: After reviewing the manuscript "Emigration and fiscal gap in a population-exporting region," I have identified several areas for improvement that span across clarity, structure, and detail. Here are my suggestions:

1. Abstract (Lines 58-72): Clarify the key findings and their implications more succinctly. Consider highlighting the novelty of your research and its contribution to the existing body of knowledge more explicitly. The current abstract presents the results well but could benefit from a more concise presentation of the study's significance.

2. Introduction (Lines 26-39): Provide a clearer statement of the research problem and its relevance in the first two paragraphs. This can help in immediately capturing the reader's interest. Also, elaborate on the theoretical and practical implications of your findings in the introduction to set the stage for your study's contributions.

3. Literature Review (Lines 40-66): Enhance the literature review by incorporating a discussion on gaps in the current research landscape that your study aims to fill. This could involve a more detailed critique of existing models and a clearer linkage between your research question and the identified gaps.

4. Theoretical Model (Lines 96-176): Increase the clarity of the model's assumptions and its applicability to the empirical context of your study. It would be beneficial to provide a more detailed rationale for the chosen model parameters and variables, ensuring the reader understands why specific assumptions were made.

5. Empirical Analysis (Lines 175-696): Improve the explanation of your methodology, particularly the choice and application of the two-step GMM model. A more detailed justification for the model selection and a clearer explanation of the steps involved in the analysis would aid reader comprehension.

6. Results and Discussion (Lines 697-728): Provide a deeper analysis of the results, focusing on the implications of your findings for policy and future research. This could include a discussion on the limitations of your study and suggestions for how future research could build upon your findings.

7. Conclusion (Lines 729-731): Strengthen the conclusion by summarizing the key findings, their implications, and potential future research directions more robustly. It would be beneficial to link back to the research problem and objectives outlined in the introduction, providing a cohesive end to the manuscript.

General Suggestions:

- Throughout the manuscript, pay attention to the clarity of your writing. Simplify complex sentences and clarify technical terms to make the manuscript accessible to a broader audience.

- Consider adding tables or figures that summarize the theoretical model and empirical findings for readers who may benefit from visual representations of complex information.

- Ensure consistency in the formatting of references, figures, and tables throughout the document.

Reviewer #2: Following are the points/quries that are needed to be considered.

1. Why and what is the reason not taking/considering up to date data for the analysis i.e.2023 year books.

2. The writers must make a proper review of the existing literature. The mere table, represented by the title “The literature review of the issue of migration and public finance represented” on page 04 in insufficient to represent proper review of literature.

3. Which correlation test is used as mentioned on page 20

4. Abstract and Conclusion seems same. Re-write both but more specifically conclusion part should show little more detail regarding the study and as the proper format of the part may be. It must incorporate the required information; which this part should include.

5. Please mention rationale/justification behind the selection of variables & analytical technique/techniques.

6. Write and properly explain separate heading showing background of the study.

6. PLOS authors have the option to publish the peer review history of their article (what does this mean?). If published, this will include your full peer review and any attached files.

Reviewer #1: **Yes: **Deni Adha Akbari

Reviewer #2: **Yes: **Dr. Zulfiqar Ali

---

## [Author Response · Author response to Decision Letter 0]

12 Apr 2024

Response to reviewers

Reviewer #1:

After reviewing the manuscript "Emigration and fiscal gap in a population-exporting region," I have identified several areas for improvement that span across clarity, structure, and detail. Here are my suggestions:

1. Comment:

Abstract (Lines 58-72): Clarify the key findings and their implications more succinctly. Consider highlighting the novelty of your research and its contribution to the existing body of knowledge more explicitly. The current abstract presents the results well but could benefit from a more concise presentation of the study's significance.

Reply:

We sincerely thank you for your valuable feedback about the novelty and significance of the research in the abstract. We have added the study's significance in revised manuscripts. The added details are as follows:

The contribution of research is twofold. On the one hand, it fills the theoretical gap between emigration and fiscal gap because previous studies have paid little attention to the fiscal problems of local government of population outflow. On the other hand, the selection of Northeast China that has been subject to long-term out-of-population migration is good evidence to verify this theory, which is tested very well using the 2S-GMM model. The comprehensive discussion on the relationship between emigration and fiscal gap is helpful to guide those continuous population-exporting regions that are facing a huge fiscal gap how to solve the fiscal gap and unsustainability from the perspective of fiscal revenue and expenditure.

2. Comment:

Introduction (Lines 26-39): Provide a clearer statement of the research problem and its relevance in the first two paragraphs. This can help in immediately capturing the reader's interest. Also, elaborate on the theoretical and practical implications of your findings in the introduction to set the stage for your study's contributions.

Reply:

We sincerely thank you for your valuable feedback that we have used to improve the quality of our manuscript. The revision is divided into two parts. One is that we have made a clearer statement of the research problem in the first two paragraphs. The other is that we have made a more in-depth elaboration of the theoretical and practical implications of the research findings. We have added the modifying contents to the manuscript and the details are as follows：

Changes to the first two paragraphs:

In developing countries, due to the different speed of economic development among cities, the income gap and living cost gap of people are constantly widening, which leads to enormous gaps of GDP among cities. And then affects the fiscal decisions of local governments and the change of specific fiscal revenue and expenditure. In a short period of time, the population-exporting regions have to face the lack of labor supply and the decline in investment because the people who are attracted to work and settle in the rapidly developing areas leave their original places of residence. More seriously, the local governments of the population-exporting regions have to face a huge fiscal gap, which is caused by insufficient fiscal revenue and huge fiscal expenditure pressure. In a large period of time, the massive emigration has led to a long-term fiscal unsustainability of the local governments who are at higher risk of developing fiscal and economic crisis. Thus, it is essential to cause alarm and attention to the population-exporting regions which is the vulnerable groups. We should build a theoretical model to explain the impact mechanism of population migration on fiscal gap of population-exporting regions and find empirical evidence for verification, which can not only supplement the theoretical system of migration and public finance, but also help governments find strategies to ensure that fiscal gap be not widened and fiscal sustainability is maintained.

About the theoretical and practical implications:

The contribution of research is twofold. On the one hand, it fills the theoretical gap between emigration and fiscal gap because previous studies have paid little attention to the fiscal problems of local government of population outflow. On the other hand, the selection of an area that has been subject to long-term out-of-population migration is good evidence to verify this theory, which is tested very well using the 2S-SGMM model. The comprehensive discussion on the relationship between emigration and fiscal gap is helpful to guide those continuous population-exporting regions that are facing a huge fiscal gap how to solve the fiscal gap problem from the perspective of fiscal revenue and expenditure.

3. Comment:

Literature Review (Lines 40-66): Enhance the literature review by incorporating a discussion on gaps in the current research landscape that your study aims to fill. This could involve a more detailed critique of existing models and a clearer linkage between your research question and the identified gaps.

Reply:

Thank you very much for your valuable comment. We have made a more detailed review and summary. This modification is reflected in the body of the paper, as follows:

Drawing from the literature review on the relationship of migration and public finance, the shortage of existing researches are as follows: First, under the topic of the relationship between migration and public finance, it is paid more attention to immigration and the financial situation of the economies that receive them. There are relatively few studies on the impact of emigration on the fiscal situation of domestic population-exporting region. We can see that people seem to miss the point that the losses suffered by the economies that send migrants are not the same as the benefits gained by the economies that receive them. This neglect has led to a blind eye to the damage of migration to the economic and fiscal balance of population-exporting region. Thus, the relationship of emigration and the fiscal situation should require in-depth scholarly attention which is exactly what this paper does. Second, most of the research have not fully discussed the fiscal part, which are usually from the perspective of the relationship between fiscal expenditure and immigration, or between fiscal policy and immigration. This paper would comprehensively analyze the relationship between the fiscal revenue and expenditure of a region of emigration, in order to fill the gap of existing research.

4. Comment:

Theoretical Model (Lines 96-176): Increase the clarity of the model's assumptions and its applicability to the empirical context of your study. It would be beneficial to provide a more detailed rationale for the chosen model parameters and variables, ensuring the reader understands why specific assumptions were made.

Reply:

We sincerely thank you for your valuable feedback that we have used to improve the quality of our manuscript. We have increased the clarity of the model's assumptions in order to apply to the empirical context of the study. The revised details are as follows:

Nevertheless, local debt does not change significantly with emigration. It is not a core explanatory variable in following models. Therefore, the Hypothesis 1 of the model is as follows:

Hypothesis 1: The balance of government debts is as a constant in the model.

Based on past experiences, there is a direct relationship between nontax revenue and tax revenue in China, that the tax revenue of local governments is getting more, while the non-tax revenue is getting less. We can identify the non-tax revenue as a part of fiscal revenue related to taxation. Therefore, the discussion of fiscal revenue only establishes the equation from the perspective of taxation and the Hypothesis 2 of the model is as follows: 

Hypothesis 2: The non-tax revenue as the part of as a part of fiscal revenue related to taxation is a constant in the model.

According to the classification of the economic nature of expenditure, the structure of fiscal expenditure that is assumed to vary automatically with GDP (Y_(i,t)) can be divided into two categories: government purchase expenditure (〖PS〗_(i,t)) and government transfer expenditure (〖TS〗_(i,t)). Furthermore, government purchasing expenditure can be subdivided into two parts, one is the expenditure to maintain normal operation of local governments, which in this paper is called 〖PS〗_(i,t)^a. The other is the expenditure closely related to people's daily production and life, such as education, environmental protection, culture, science and technology, which is called purchase spending (〖PS〗_(i,t)^b). This type of the spending is assumed to vary automatically with the fiscal policy (P_t^d) related to public demand. Government transfer expenditure is assumed to vary automatically with the fiscal policy (P_t^s) related to the local fiscal policy, which is including social security and employment expenditures, and health care expenditures. The Hypothesis 3 of the model is as follows:

Hypothesis 3: The division of government expenditure types is related to the fiscal objectives achieved by the local governments.

5. Comment:

Empirical Analysis (Lines 175-696): Improve the explanation of your methodology, particularly the choice and application of the two-step GMM model. A more detailed justification for the model selection and a clearer explanation of the steps involved in the analysis would aid reader comprehension.

Reply:

We sincerely thank you for your valuable feedback. We have a more detailed explanation of the two-step GMM model selection in the revised S1 Appendix. The reasons for the choice and application of two-step GMM are as follows:

The dynamic panel model included three methods, which were namely difference GMM that all possible lag variables were the instrumental variables (Arellano and Bond 1991) [S1], namely level GMM that the instrumental variables were not related to the composite perturbation term of the horizontal equation (Arellano and Bover 1995) [S2], and namely system GMM that combined the above two methods together for GMM estimation (Blundell and Bond 1998) [S3]. In the empirical analysis, the choice of model was considered from two aspects. On the one hand, the time dimension was small and the individual cross-section dimension was large in the panel data. On the other hand, the multiple test results were judged after regression. Eventually, system GMM was chosen to achieve the empirical analysis. The dynamic panel GMM estimation can be divided into one-step estimation and two-step estimation according to the different weight matrices. The two-step system GMM method is chosen in order to ensure the satisfaction of three conditions. First, the standard error presents obvious decreasing bias. Second, the perturbation term is not autocorrelated. Third, the moment condition is not overly constrained. Meanwhile, in order to ensure the effectiveness of system GMM, two tests are selected to identify the effectiveness of the instrumental variables and the estimated results. 

[S1] Arellano, M. and Bond, S. 1991. “Some Tests of Specification for Panel Data: Monte Carlo Evidence and an Application to Employment Equations.” Review of Economic Studies 58(2): 277–297. https://doi.org/10.2307/2297968.

[S2] Arellano, M. and Bover, O. 1995 “Another Look at the Instrumental Variable Estimation of Error-Components Models.” Journal of Econometrics 68: 29–51. https://doi.org/10.1016/0304-4076(94)01642-D.

[S3] Blundell, R. and Bond, S. 1998. “Initial Conditions and Moment Restrictions in Dynamic Panel Data Models.” Journal of Econometrics 87(1): 115–143. https://doi.org/10.1016/j.jeconom.2023.03.001.

6. Comment:

Results and Discussion (Lines 697-728): Provide a deeper analysis of the results, focusing on the implications of your findings for policy and future research. This could include a discussion on the limitations of your study and suggestions for how future research could build upon your findings.

Reply:

We sincerely thank you for your valuable feedback that we have used to improve the quality of our manuscript. The details of the changes have been presented in the corresponding section of the revised manuscript. The added analysis are as follows:

A deeper analysis of the fiscal revenue gap：

Under the current tax rate and tax base, the consequence of emigration would continue to worsen the situation of local fiscal revenue. It has to add some new ways if the local governments want to increase the local fiscal revenue. In the past 10 years, many local governments in Northeast China believed that the income from land usage right and taxes brought by the real estate industry is the effective way to maintain local fiscal revenue. However, with the accumulation of bubbles in the real estate industry, it is difficult for this way to have positive effects in the future.

A deeper analysis of the fiscal expenditure gap：

In the short run, there has been no significant reduction in government spending to offset the decline in fiscal revenue. it would become more serious in the long run if this situation is not adjusted in the government budget. The fiscal sustainability will continue to deteriorate.

In this research, we are limited to the discussion of purchase spending and transfer expenditure. If possible, we should go deep into each sub-report and item of the government budget to discuss whether it can be reduced or adjusted to optimize the fiscal expenditure structure in the future.

7. Comment:

Conclusion (Lines 729-731): Strengthen the conclusion by summarizing the key findings, their implications, and potential future research directions more robustly. It would be beneficial to link back to the research problem and objectives outlined in the introduction, providing a cohesive end to the manuscript.

Reply:

We sincerely thank you for your valuable feedback that we have used to improve the quality of our manuscript. We rewrote our conclusions and put it into our manuscript. The details are as follows:

In conclusion, the following conclusions are obtained. We have explored the relationship between emigration and fiscal gap from the general equilibrium model. It needs to be investigated the changes in the fiscal gap over time. On the one hand, In the short and long term, emigration can expand the fiscal revenue gap through the superimposed effect of tax rate and tax base. It has to find some new ways if the local governments want to increase the local fiscal revenue, such as introducing new local taxes or non-tax revenue. On the other hand, the exodus widens the fiscal spending gap in the short term while the spending gap would continue to widen in the long run if the local governments do not change their strategies. In the structure of fiscal expenditure, the government purchasing expenditure is mainly invested in the early stage and maintained in the later stage so that it is difficult to really reduce the financial burden of the government on emigration in the short term. Meanwhile, in order to retain population and attract new ones, the scale of purchase spending based on GDP would increase. Due to the structure of local population, the proportion of transfer expenditure would be relatively high. In the empirical analysis, we have taken 36 cities in Northeast China as the research samples, which have faced the serious population emigration in recent ten years, to verify the general equilibrium model. The emigration has a positive impact on the fiscal gap, that the more severe population emigration, the larger the fiscal gap. When the trend of emigration becomes irreversible, the subsequent efforts of local governments to expand fiscal expenditure for attraction population would not only fail to revive the regional economy, but aggravate the expansion of fiscal gap.

This work fills the theoretical gap between migration and public finance, and its conclusions can help ameliorate fiscal unsustainability. Future studies can continue to explore the economic gains of population-receiving areas and the economic losses of population-exporting areas. We can find a suitable combination of fiscal policies to reduce the efficiency losses. We can consider that the fiscal balance of these population-exporting regions can be better solved from the perspective of space.

General Suggestions:

- Throughout th

---

## [Editor Report · Decision Letter 1]

16 Apr 2024

Emigration and fiscal gap in population-exporting region

PONE-D-24-02684R1

Dear Dr. Li

We’re pleased to inform you that your manuscript has been judged scientifically suitable for publication and will be formally accepted for publication once it meets all outstanding technical requirements.

Kind regards,

Ghulam Rasool Madni, Ph.D

Academic Editor

PLOS ONE
---

## [Editor Report · Acceptance letter]

26 Apr 2024

PONE-D-24-02684R1 

PLOS ONE

Dear Dr. Li, 

I'm pleased to inform you that your manuscript has been deemed suitable for publication in PLOS ONE. Congratulations! Your manuscript is now being handed over to our production team.

Kind regards, 

on behalf of

Dr. Ghulam Rasool Madni 

Academic Editor

PLOS ONE